# Breast Cancer Incidence among Female Workers by Different Occupations and Industries: A Longitudinal Population-Based Matched Case–Control Study in Taiwan

**DOI:** 10.3390/ijerph191610352

**Published:** 2022-08-19

**Authors:** Cheng-Ting Shen, Hui-Min Hsieh, Yun-Shiuan Chuang, Chih-Hong Pan, Ming-Tsang Wu

**Affiliations:** 1Department of Family Medicine, Kaohsiung Municipal Ta-Tung Hospital, Kaohsiung Medical University Hospital, Kaohsiung Medical University, Kaohsiung City 807, Taiwan; 2Department of Public Health, Kaohsiung Medical University, Kaohsiung City 807, Taiwan; 3Department of Medical Research, Kaohsiung Medical University Hospital, Kaohsiung City 807, Taiwan; 4Department of Community Medicine, Kaohsiung Medical University Hospital, Kaohsiung City 807, Taiwan; 5Center for Big Data Research, Kaohsiung Medical University, Kaohsiung City 807, Taiwan; 6Research Center for Precision Environmental Medicine, Kaohsiung Medical University, Kaohsiung City 807, Taiwan; 7Department of Family Medicine, Kaohsiung Medical University Hospital, Kaohsiung City 807, Taiwan; 8Institute of Labor, Occupational Safety and Health, Ministry of Labor, Taipei 40767, Taiwan; 9Department of Public Health and PhD Program in Environmental and Occupational Medicine, Kaohsiung Medical University, Kaohsiung City 807, Taiwan

**Keywords:** breast cancer risks, female workers, occupational industries

## Abstract

Background: Breast cancer is the leading cause of cancer incidence worldwide and in Taiwan. The relationship between breast cancer and occupational types remains unclear. This study aimed to investigate lifetime breast cancer incidence by different occupational industries among female workers in Taiwan. Methods: A population-based retrospective case–control study was conducted using three nationwide population-based databases. Matched case and control groups were identified with 1-to-4 exact matching among 103,047 female workers with breast cancer diagnosed in 2008–2017 and those without breast cancer. Their lifetime labor enrollment records were tracked using the National Labor Insurance Database, 1950–2017. Conditional logistic regression was used to analyze the association between types of occupational industries and risk of incident breast cancer. Results: Our study found slightly significant breast cancer risk among the following major occupational classifications: manufacturing (OR: 1.027, 95% CI: 1.011–1.043); wholesale and retail trade (OR: 1.068, 95% CI: 1.052–1.084); information and communication (OR: 1.074, 95% CI: 1.043–1.105); financial and insurance activities (OR: 1.109, 95% CI: 1.086–1.133); real estate activities (OR: 1.050, 95% CI: 1.016–1.085); professional, scientific, and technical activities (OR: 1.118, 95% CI: 1.091–1.145); public administration, defense, and social security (OR: 1.054, 95% CI: 1.023–1.087), education (OR: 1.199, 95% CI: 1.168–1.230); and human health and social work activities (OR: 1.125, 95% CI: 1.096–1.156). Conclusions: Greater percentages of industrial occupations (i.e., manufacturing, wholesale and retail, or health professionals) were associated with slightly increased breast cancer risk. Further studies should investigate the possible risk factors among female workers in those industries with slightly higher incidence of breast cancer.

## 1. Introduction

The GLOBOCAN 2020 estimate of 2.3 million new female breast cancer cases exceeds lung cancer as the top cause of cancer incidence worldwide [1]. Breast cancer also became the leading cause of overall cancer incidence in Taiwan [2]. The higher incidence rate may be associated with known risk factors, such as genetic mutations, family history, reproductive and hormonal risk factors (early menarche, late menopause, late first-birth age, less childbirth, less breastfeeding, hormone therapy), and lifestyle risk factors (alcohol consumption, being overweight or obese after menopause), and may reflect increased cases due to organized or opportunistic mammography screening [1]. However, occupational contributions to breast cancer risk factors are not completely understood.

The International Agency for Research on Cancer (IARC) noted several factors with sufficient or limited evidence for human breast cancer risk, including X- and gamma-radiation, night-shift work, dieldrin, ethylene oxide, and polychlorinated biphenyls. The relationship between night-shift work, defined as work at the usual sleeping time and including trans-meridian air travel, and breast cancer incidence had been investigated in past studies among nurses [3,4]. In addition, some chemical substances, known as endocrine-disrupting chemicals, including ethylene oxide, polycyclic aromatic hydrocarbons, perfluorooctanoic acid, and several pesticides, may change the normal endocrine processes and the development of mammary glands [5]. Sedentary work and low physical activity, which may contribute to increasing estrogen levels during the menstrual cycle, may be breast cancer risk factors [6]. A Japanese cohort study among 19,041 female workers found higher breast cancer risk in office workers, and in those who work in the sitting position and walk less daily [6].

Frequently discussed occupations with higher breast cancer risk include flight attendants, medical professionals, and workers in some production positions. A meta-analysis found a combined standard incidence ratio of breast cancer among female flight attendants of 1.40 (95% CI: 1.30–1.50) [7], and a study evaluating cancer prevalence between flight attendants and the general population found higher breast cancer prevalence ratios (1.5, 95% CI: 1.02–2.24) among female flight attendants [8]. In addition, Kjaer et al. (2009) focused on 92,140 female Danish nurses and found significantly elevated breast cancer risk [9]. Wegrzyn et al. (2017) found that higher breast cancer risk was associated with a longer working year or more cumulative shift work in the Nurses’ Health Study II [4].

Regarding breast cancer risk among other occupations, a Canadian case–control study found that automotive plastics manufacturing, food canning, and metalworking were associated with higher breast cancer risk [10]. A CECILE study conducted in France found marginally elevated breast cancer risk in some occupations, including several white-collar occupations, rubber and plastics production, textile production, nursing, and tailoring/dressmaking [11]. Another cohort study conducted in Canada using the occupational disease surveillance system followed up from 1983 to 2016 and found higher breast cancer risk among female workers in occupations such as management, administrative roles, clerical roles, medicine and health, and teaching [12].

Most previous studies were conducted in Western countries, with fewer population-based studies focused on Asia. For instance, a population-based study among 74,942 female workers in Shanghai found higher breast cancer risk among white-collar and several types of production workers [13]. A Japanese case–control study found reduced breast cancer risk among female workers in high physical-activity jobs, including agriculture, manufacturing, construction and mining, and transport, compared with sales [14]. A longitudinal nationwide analysis followed up for longer than 10 years in Korea found lower breast cancer risk among lower socioeconomic occupations such as blue-collar workers, service workers, and sales workers [15]. In Taiwan, studies focused on breast cancer risk among different occupations remain limited, especially studies with longer follow-up time. This study aimed to investigate breast cancer incidence among female workers from different occupations in Taiwan.

## 2. Methods

### 2.1. Study Design and Data Source

This study conducted a retrospective case–control study design among women with an incident breast cancer diagnosis between 2008 and 2017, matched with those with no breast cancer diagnosis. Using three nationwide population-based labor insurance databases in Taiwan, we then tracked the study cohorts’ labor enrollment records for almost 70 years from 1950 to 2017 to explore potential breast cancer risks in different occupational industries over a lifetime period. The first was the National Labor Insurance Database (NLID) from the Ministry of Labor, containing representative data on labor enrollment profiles from 1950 to 2017. According to the official Labor Insurance Act in Taiwan, all industrial workers and companies are required to enroll in National Labor Insurance to protect workers’ livelihoods and promote social security. The NLID provided detailed information regarding labor enrollment profiles, company profiles, and industrial classifications. The industrial classifications were based on the tenth version of the standard industrial classification of Taiwan’s National Statistics. The second database was the National Taiwan Cancer Registry, containing records of all cancer diagnoses, tracked from 1979 to 2017. The third was a National Death Registry tracked from 1971 to 2017, containing accurate death dates and causes. These databases are complete and representative of all Taiwanese workers, and were linked using encrypted identifiers. All data analyses were completed in Taiwan’s Health and Welfare Data Science Center, Ministry of Health and Welfare, in 2020–2021. This study was approved by the Institutional Review Board.

### 2.2. Study Population

We first identified female workers newly diagnosed with breast cancer as a case group using the National Taiwan Registry Database between 2008 and 2017 (*n* = 124,007), linked with the National Death Registry and the NLID. We then excluded those aged younger than 20 years at incident breast cancer diagnosis, those with any cancer diagnosis or death records prior to the date of newly diagnosed breast cancer, those with no labor insurance enrollment records prior to the date of newly diagnosed breast cancer, those with missing specific working company records in the NLID, and those with total cumulative job tenure less than 3 months in working industries. The index date for the case group was defined as the date of incident breast cancer diagnosis, and the index date was assigned to the comparison group matched based on age and major working locations. For the comparison group, we identified female workers with no breast cancer diagnosis and enrolled in the NILD with at least 3 months of cumulative job tenure in working industries between 1950 and 2017. We further excluded comparison subjects with any cancer diagnoses or death records prior to the study end date (31 December 2017). Finally, 103,051 female workers with incident breast cancer (case group) and 8,111,155 female workers without breast cancer (comparison group) were included in the analysis. 

To determine adequate comparisons with the same distribution of age and major working locations, we used a 1-to-4 exact matching approach to find matched comparison groups. Final matched case and comparison groups included 103,047 female workers with incident breast cancer and 412,188 female workers without breast cancer. Figure 1 presents a detailed flow chart of inclusion and exclusion criteria. 

### 2.3. Measurements

The first diagnosis in the National Cancer Registry after the index date with ICD-9-CM diagnosis code 174 or ICD-10-CM diagnosis code C50 was identified as incident breast cancer among female workers between 2008 and 2017. The key exposure variables were occupational industries worked in prior to the incident breast cancer diagnosis (index date) among case and comparison female workers based on the National Labor Insurance Database enrollment records. Specifically, we tracked back to investigate potential breast cancer incident risks in different occupational industries over a lifetime period using the national labor enrollment records from 1950 to 2017 in Taiwan. The industrial classifications were based on the tenth version of the standard industrial classification of Taiwan’s National Statistics, at four levels: section, division, group, and class. We focused on analyzing the association between occupational industries at the section (from section A to S) and division (from division 1 to 96) levels, and the incident breast cancer diagnosis.

### 2.4. Statistical Analysis

A chi-square test was used to evaluate categorical variables between the case and control groups. The association between types of occupational industries and the risk of incident breast cancer was analyzed by using conditional logistic regression, while controlling for total cumulative job tenure years. Odds ratios (ORs) and 95% confidence intervals (CIs) showed the risk of incident breast cancer. The Bonferroni and false discovery rate approaches were used to calculate adjusted *p* values among multiple specific occupational industry comparisons to handle multiple testing concerns [16]. The data analysis was generated using SAS^®^ software, Version 9.4 of the SAS System for Windows (SAS institute, Cary, NC, USA). A *p* value < 0.05 was considered statistically significant.

## 3. Results

Table 1 summarizes demographic characteristics, working locations, and total cumulative job tenure between matched case and control female workers based on their index age and working locations. Mean index ages for matched cohorts were 53.21 years, and greater proportions worked in the major working locations of Taipei, Taichung, and Kaohsiung. Mean total cumulative job tenures were 15.97 years among women with incident breast cancer. The percentages of women in every job tenure in years were 3.49% for less than 1 year, 6.79% for 1–3 year, 6.43% for 3–5 years, 15.56% for 5–10 years, 29.92% for 10–20 years and 37.81% for over 20 years.

Table 2 provides ORs and multiple testing results by the standard industrial classification system at the section levels. Compared with comparison groups at the section level, significantly greater percentages of case groups (patients with breast cancer) worked in manufacturing (section C; 69.85%, OR 1.027, *p* < 0.0002); wholesale and retail trade (section G; 38.63%, OR 1.068, *p* < 0.0001); information and communications (section J; 6.61%, OR 1.074, *p* < 0.0001); financial and insurance activity (section K; 12.74%, OR 1.109, *p* < 0.0001); real estate activity (section L; 4.77%, OR 1.050, *p* < 0.0039); professional, scientific, and technical activities (section M; 9.87%, OR 1.118, *p* < 0.0001); public administration and defense, and social security (section O; 5.39%, OR 1.054, *p* < 0.0001); education (section P; 8.55%, OR 1.199, *p* < 0.0001); and human health and social work activities (section Q; 7.76%, OR 1.125, *p* < 0.0001). This study also provides detailed results at the division level in Appendix A, Table A1.

## 4. Discussion

We conducted a population-based retrospective case–control study and tracked back the national labor enrollment records for approximately 70 years from 1950 to 2017 to investigate potential breast cancer incident risks in different occupational industries over a lifetime period among female workers. Our study found significant breast cancer risk among the major occupational classifications including manufacturing; wholesale and retail trade; information and communication; financial and insurance activities; real estate activities; professional, scientific and technical activities; public administration; defense; social security; education; and human health and social work activities.

Some of our findings were consistent with those from previous population-based international research investigating the relationship between breast cancer and kinds of occupations [13,17]. For example, Ji et al. (2008) conducted a population-based cohort study in Shanghai and reported higher breast cancer risk associated with several production industries, postal/telecommunication workers and white-collar professionals [13]. Pukkala et al. (2009) investigated the relationship between occupation and cancer among 15 million people in five Nordic countries and found higher breast cancer risk among jobs such as military personnel, dentists, physicians, technical workers, administrators, clerical workers, teachers, artistic workers, religious workers, sales agents, transport workers, and postal workers [17].

Our findings suggest higher breast cancer risk for female workers in most subgroups of manufacturing with possible exposure to several kinds of organic solvents. Past studies found similar results. For example, Oddone et al. (2013) reported a case–control study in Italy that found that breast cancer incidence was associated with employment in textile, rubber, paper, and electrical manufacturing industries [18]. Kaneko et al. (2020) conducted a case–control study investigating cancer risk among different manufacturing industries and found higher breast cancer risk among industries including clothing, chemicals, non-ferrous metals and products, and communication electronics, compared with a food manufacturing group [19]. Another study conducted in Canada found elevated breast cancer risk among female workers in plastics and rubber product fabricating industries [20]. An integrated review in 2021 found that workplace exposures such as cadmium, radiation, night work, and some chemical products including pesticides, solvents, and alkylphenols were associated with breast cancer incidence [21]. Alkylphenolic compounds, which are produced as organic chemicals including non-ionic surfactants, may come into contact in the workplace during the production process and expose workers using industrial detergents, pesticides, cosmetics and hair dyes, and specialty paints; these are considered endocrine disruptors with an effect on estrogen receptors’ mediation [22]. Organic solvents, widely used in manufacturing industries including electronics, semiconductors, reinforced plastics, paint, dry cleaning, textiles, and leather, are highly lipophilic and, stored in the breast adipose tissue, could promote carcinogenesis [23]. A system review and meta-analysis found higher breast cancer risk among workers exposed to organic solvents compared with no exposure (OR 1.18, 95% CI 1.11–1.25) [23]. Erika et al. (2017) found higher breast cancer risk among female autoworkers exposed to metalworking fluid from automobile manufacturing plants in Michigan (HR:1.13, 95%CI: 1.03–1.23) [24].

Results in the existing literature are mixed with respect to the association between agricultural work and breast cancer risk. Agricultural workers may be exposed to chemicals, pesticides, solvents, and biological agents [14]. An agricultural Health Study discussed the relation between insecticide and breast cancer incidence in farmers’ wives and found higher breast cancer risk in women who had ever used several organophosphate insecticides, but results were not consistent regarding use of insecticides between farmers and their wives [25]. A study aimed at cancer risk among 70,570 agricultural workers in Canada found no significant reduced breast cancer risk among female workers [26]. Togawa et al. (2021) conducted an international consortium study (AGRICOH) using eight prospective agricultural worker cohorts and found reduced breast cancer risk among women, which may be associated with high physical activity [27]. Our ecological study also found lower breast cancer risk among workers in agriculture and animal husbandry. 

We found higher breast cancer risk among telecommunications workers, which may be partly associated with exposure to electromagnetic fields (EMF). Several occupations may be with higher workplace EMF exposure, including telecommunications, broadcasting, and security and identification. Kliukiene et al. (2003) reported a cohort study that found elevated breast cancer risk among female Norwegian radio and telegraph workers, and EMF exposure was associated with different subtypes of breast cancer among women in different age groups [28]. McElroy et al. (2007) conducted a population-based case–control study in the United States and found that women with different levels of occupational EMF exposure had slightly higher breast cancer risk, but results lacked statistical significance [29]. However, a Swedish case–control study using a population-based registry found no association between occupational EMF exposure and breast cancer risk [30]. Although EMF was thought to interfere with melatonin synthesis and increase estrogen level, which is related to breast tissue proliferation, further studies are necessary to investigate the relationship between EMF exposure and cancers [5].

We found elevated breast cancer risk among female water transportation and air transportation workers, and marginal risk among land transportation workers. A possible exposure may be exhaust from engines powered by diesel and other fuels, composed of several gases including carbon monoxide, nitrogen oxides, and polycyclic aromatic hydrocarbons [31]. However, a population-based case–control study in Australia that used questionnaires and telephone interviews found no association between breast cancer risk and engine exhausts from occupational exposure [32]. Another case–control study in Denmark of 38,375 women diagnosed with breast cancer found that diesel exhaust was related with estrogen receptor-negative breast tumors among women aged younger than 50 years (OR 1.26, 95% CI 1.09–1.46) [33].

Another possible exposure may be night shift work, which is important in some public service occupations, such as transportation, electricity, and health care [34]. The International Agency for Research on Cancer had classified night work as a Group 2A carcinogen in 2007, as disruption in circadian rhythm with interfering melatonin synthesis would influence sex hormone production and normal cell proliferation [21,34]. Menegaux et al. (2013) conducted the CECILE study in France including women working night shifts in occupations such as hotels and restaurants, transportation and communication, health and social work, and some manufacturing. They found that working overnight shifts or night work for longer than 4.5 years was related to incident breast cancer cases, particularly among women who began night work before their first full-term pregnancy [35].

Sales, retail, and accommodation service workers had higher breast cancer risk in our study. Higher breast cancer incidence has been observed in past studies [11]. Our study also found higher breast cancer incidence in other white-collar occupations, such as publishing, media-related activities, financial and insurance activities, real estate, advertising and market research, and specialized design activities. The potential exposure from the environment in these occupations is not obvious, and the possible risk may be their sedentary nature. For example, a study examined 19,041 Japanese female workers and found elevated breast cancer risk among office workers and women working in the sitting position, after adjusting for other covariates, including reproductive-related factors [6]. Another systematic review and meta-analysis found an excess 15.5% risk of breast cancer associated with occupations involving sedentary work [36].

We observed higher breast cancer risk in some white-collar occupations, such as legal and accounting activities, education, and medicine, possibly partly associated with reproductive risk factors including older age at first child’s birth and lower parity due to longer training time in highly professional field [37,38]. One recent population-based cohort study among female health professionals in Taiwan with a 35-year follow-up found elevated incident breast cancer among health professionals overall and among several types of professionals including physicians, registered nurses, pharmacists, medical technologists, and psychologists; possible occupational exposures included night shift work, ionizing radiation, work stress, and some chemicals from hospitals [39]. Several previous studies have reported the relation between night shift work and incident breast cancer among nurses. In addition, hospital physicians may work overtime, and the prevalence of job stress and burnout has been reported to be higher in nurses and physician assistants among hospital employees [3,4,34]. Heikkilä et al. (2013) conducted a meta-analysis focused on 116,000 European women and found that job strain was insignificantly associated with elevated risk for breast, colorectal, lung, or prostate cancers [40]. A multi-cohort study of the relationship between long working hours and cancer risk found that working more than 55 h per week was associated with elevated breast cancer risk [41].

The strength of the current case–control study is that we designed a longitudinal cohort combining three nationwide population-based databases, creating a large sample size with the labor insurance registry from 1950 to 2017 and exact matching 1-to-4. Each section and division of occupational industries and job tenure was collected from the National Labor Insurance enrollment databases, possibly reducing recall bias. However, our study had some limitations. First, several unobservable risk factors could be confounders because of secondary data analysis, including hormone-related risk factors, family history, alcohol consumption, and possible occupational exposures such as night shift work, organic chemicals, physical activities, working hours and even different exposure and its duration in each section and division level of the industry. We also lack the records on labors’ attendance including information on sick leaves for maternity or pregnancies in each industry. Second, some subgroup occupations with higher breast cancer risk may be overestimated due to smaller sample sizes. Third, the total cumulative job tenures were confounding factors for conditional logistic regression in our study. However, we did not particularly investigate the association between breast cancer incidence and job employment duration of each industry for individual women. Further study may investigate the relationship between breast cancer incidence and each job employment duration in each industry. Fourth, mammography screening utilization and screening adherence might also be a potential confounder while investigating breast cancer risks among these occupational industries. However, the current study did not examine mammography rate among occupational industries. Future study may further investigate whether mammography rate varies with occupations. Finally, our findings may not generalize to other countries because the occupational exposures may be at different levels in each country.

## 5. Conclusions

Our study found slightly significant breast cancer risk among the major classification of occupations including manufacturing, wholesale and retail trade, information and communication activities, financial and insurance activities, real estate activities, professions, scientific and technical activities, public administration, defense, social security, education, and human health and social work activities from the section and division levels according to the standard industrial classification of Taiwan’s National Statistics. Further studies should investigate the possible risk factors among female workers in those industries with slightly higher incident breast cancer.

## Figures and Tables

**Figure 1 ijerph-19-10352-f001:**
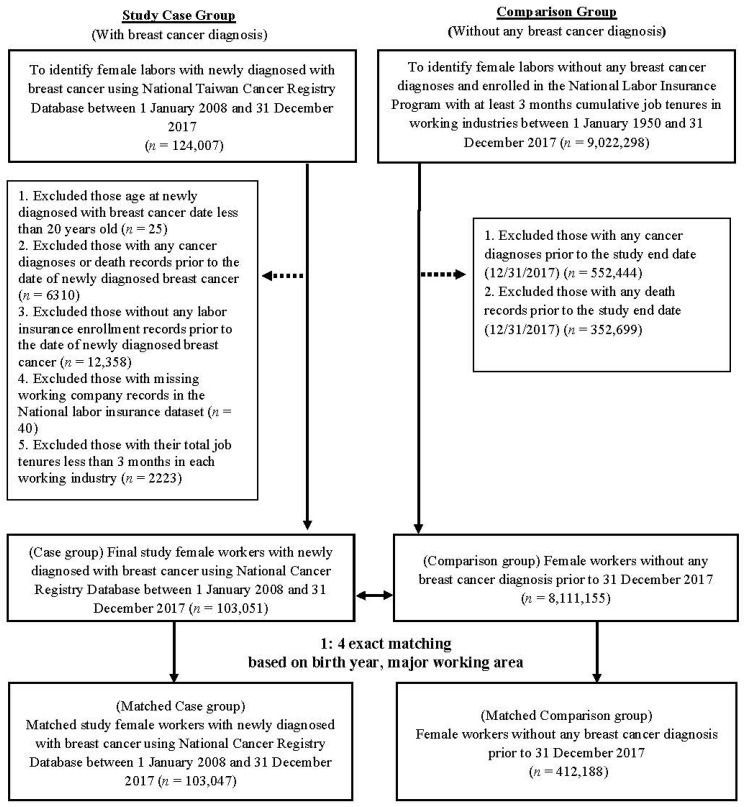
Flow chart of sample inclusion and exclusion criteria in this study.

**Table 1 ijerph-19-10352-t001:** Demographic and working characteristics in 1:4 matched case and comparison female workers in Taiwan.

	Case Group (with Breast Cancer)	Comparison Group (without Breast Cancer)	*p* Value
*n*	103,047	412,188	
Age in years (mean ± SD) *	53.21 (±10.84)	53.21 (±10.84)	
Age categories (*n*, %)			
<44	21,083 (20.46%)	84,332 (20.46%)	1.000
45–54	38,195 (37.07%)	152,780 (37.07%)	
55–64	28,257 (27.42%)	113,028 (27.42%)	
65–74	11,577 (11.23%)	46,308 (11.23%)	
75+	3935 (3.82%)	15,740 (3.82%)	
Major working locations (*n*, %) *			
Taipei	28,780 (27.93%)	115,120 (27.93%)	1.000
Keelung	1722 (1.67%)	6888 (1.67%)	
New Taipei	14,660 (14.23%)	58,640 (14.23%)	
Yilan	1853 (1.80%)	7412 (1.80%)	
Hsinchu	3595 (3.49%)	14,380 (3.49%)	
Taoyuan	7011 (6.80%)	28,044 (6.80%)	
Miaoli	1671 (1.62%)	6684 (1.62%)	
Taichung	10,598 (10.28%)	42,392 (10.28%)	
Changhua	3944 (3.83%)	15,776 (3.83%)	
Nantou	1507 (1.46%)	6028 (1.46%)	
Chiayi	2416 (2.34%)	9664 (2.34%)	
Yunlin	1793 (1.74%)	7172 (1.74%)	
Tainan	7155 (6.94%)	28,620 (6.94%)	
Kaohsiung	12,299 (11.94%)	49,196 (11.94%)	
Pingtung	2208 (2.14%)	8832 (2.14%)	
Taitung	539 (0.52%)	2156 (0.52%)	
Hualien	1122 (1.09%)	4488 (1.09%)	
Jinmen, Lienchiang county	174 (0.17%)	696 (0.17%)	
Total cumulative job tenures in year (mean ± SD)	15.97 (±9.56)	15.24 (±9.58)	<0.001
Total cumulative job tenures categories in years (*n*, %)			
<1	3596 (3.49%)	21,099 (5.12%)	<0.001
1–3	6997 (6.79%)	32,073 (7.78%)	
3–5	6624 (6.43%)	29,090 (7.06%)	
5–10	16,036 (15.56%)	66,382 (16.10%)	
10–20	30,836 (29.92%)	119,772 (29.06%)	
20+	38,958 (37.81%)	143,772 (34.88%)	

Note: SD, standard deviation. * 1 to 4 exact matching method was used to match case and control study samples by using index age and major work locations.

**Table 2 ijerph-19-10352-t002:** Odds ratios and multiple testing results by standard industrial classification system at the section level.

Section	Standard Industrial Classification System	Case Group (with Breast Cancer)	Comparison Group (without Breast Cancer)	Conditional Logistic Regression Model Results ^#^	Multiple Testing		
		*n* (%)	*n* (%)	OR (95%CI)	*p* Value	Bonferroni	FDR
	*n*	103,047	412,188				
A	Agriculture, Forestry, Fishing and Animal Husbandry	4898 (4.75%)	21,552 (5.23%)	0.877 (0.849, 0.906)	<0.0001	0.0019	0.0002
B	Mining and Quarrying	473 (0.46%)	2243 (0.54%)	0.826 (0.747, 0.913)	0.0002	0.0038	0.0004
C	Manufacturing	71,979 (69.85%)	287,009 (69.63%)	1.027 (1.011, 1.043)	0.0002	0.0038	0.0004
D	Electricity and Gas Supply	332 (0.32%)	1197 (0.29%)	1.081 (0.957, 1.222)	0.2101	1.000	0.2495
E	Water Supply and Remediation Activities	1422 (1.38%)	6093 (1.48%)	0.906 (0.854, 0.960)	0.0009	0.0009	0.0014
F	Construction	13,311 (12.92%)	54,311 (13.18%)	0.944 (0.925, 0.964)	<0.0001	0.0019	0.0002
G	Wholesale and Retail Trade	39,809 (38.63%)	149,622 (36.30%)	1.068 (1.052, 1.084)	<0.0001	0.0019	0.0002
H	Transportation and Storage	9389 (9.11%)	36,415 (8.83%)	1.001 (0.977, 1.025)	0.9564	1.0000	0.9564
I	Accommodation and Food Service Activities	10,416 (10.11%)	41,999 (10.19%)	0.963 (0.941, 0.985)	0.0012	0.0228	0.0018
J	Information and Communication	6815 (6.61%)	24,206 (5.87%)	1.074 (1.043, 1.105)	<0.0001	0.0019	0.0002
K	Financial and Insurance Activities	13,124 (12.74%)	46,731 (11.34%)	1.109 (1.086, 1.133)	<0.0001	0.0019	0.0002
L	Real Estate Activities	4914 (4.77%)	17,888 (4.34%)	1.050 (1.016, 1.085)	0.0039	0.0741	0.0053
M	Professional, Scientific and Technical Activities	10,175 (9.87%)	35,368 (8.58%)	1.118 (1.091, 1.145)	<0.0001	0.0019	0.0002
N	Support Service Activities	7838 (7.61%)	29,744 (7.22%)	1.013 (0.986, 1.040)	0.3437	1.0000	0.3628
O	Public Administration and Defense; Compulsory Social Security	5670 (5.50%)	20,910 (5.07%)	1.054 (1.023, 1.087)	0.0007	0.0133	0.0013
P	Education	8807 (8.55%)	29,385 (7.13%)	1.199 (1.168, 1.230)	<0.0001	0.0019	0.0002
Q	Human Health and Social Work Activities	8000 (7.76%)	27,826 (6.75%)	1.125 (1.096, 1.156)	<0.0001	0.0019	0.0002
R	Arts, Entertainment and Recreation	3136 (3.04%)	11,905 (2.89%)	1.024 (0.983, 1.066)	0.2535	1.0000	0.2833
S	Other Service Activities	15,782 (15.32%)	61,632 (14.95%)	0.985 (0.966, 1.005)	0.1334	1.0000	0.1690

Note: Individual major industry categories were not mutually exclusive. ^#^ Multivariable conditional logistic regression models were used to generate adjusted odds ratio, while controlling for individual total cumulative job tenures.

## Data Availability

The data that support the findings of this study are available from Taiwan’s Health and Welfare Data Science Center, Ministry of Health and Welfare, but restrictions apply to the availability of these data, which were used under license for the current study, and so are not publicly available. Data are, however, available from the authors upon reasonable request and with permission of Taiwan’s Health and Welfare Data Science Center, Ministry of Health and Welfare.

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
