# Peer review of "Breast Cancer Incidence among Female Workers by Different Occupations and Industries: A Longitudinal Population-Based Matched Case–Control Study in Taiwan"

_ijerph, 2022, doi:10.3390/ijerph191610352_

Round 1
Reviewer 1 Report
This is an interesting paper and a useful epidemiological resource focussing on possible associations between Asian occupational groups and breast cancer. Nevertheless, it is also quite a confusing read. My main concerns are the conflation of breast cancer risk and incidence which are, arguably, distinct. Breast cancer risk implies some form of risk assessment, or prevalence of risk factors, whereas the focus of your research is breast cancer incidence. Consequently, a more appropriate title would be “Breast Cancer incidence among female workers by different occupations and industries”.
In the introduction, you cite 3 possible explanations for the relationship between BC and occupational categories:
· Chemical exposure
· Shift work (disrupting diurnal rhythm)
· Sedentary occupations.
You also follow these up when you are discussing your results. However, you also throw in a number of other factors that are unsubstantiated as you did not control for these.
A notable example is 14: 294-296 where you claim that: “ higher breast cancer risk in some white-collar occupations, such as legal and accounting activities, education, and medicine, possibly partly associated with reproductive risk factors including older age at first childbirth and less parity due to longer training time”. This needs further justification with reference to related research.
Your findings are not always effectively related to other research studies. For instance, 12:202-204 you cite Ji et al., 2008 but don’t relate this to your findings. Sometimes your extensive citation of other studies reads more like a list rather than as part of an interpretative analysis. It may be an idea to consider just focussing on Asian studies with a separate section to cover how these findings compare with Westernised evidence for an association between occupational sub-group and breast cancer incidence. You may also want to think about tabulating some of the known occupational risk factors as this may break up the density of your background text.
The claim in your conclusion that the occupational groups with highest levels of BC would benefit from education about early detection is not appropriate or sufficiently justified and referenced. Do you mean education to enhance awareness of early detection (i.e., symptom awareness, BSE/mammography)? / Are you referring to the need for awareness campaigns about possible risk factors associated with certain occupations? In either case, you need to contextualise this conclusion in more detail throughout the rest of the paper. As it stands, it doesn’t seem to be a logically informed or clearly expressed conclusion to the paper.
Should the authors consider any of these points appropriate, I would be happy to review this paper again.
Author Response
Point#1: My main concerns are the conflation of breast cancer risk and incidence which are, arguably, distinct. Breast cancer risk implies some form of risk assessment, or prevalence of risk factors, whereas the focus of your research is breast cancer incidence. Consequently, a more appropriate title would be “Breast Cancer incidence among female workers by different occupations and industries”.
Our response: We appreciate the Reviewer’s suggestion. We agreed and revised the title of the manuscript accordingly as follows:
“Breast Cancer Incidence among Female Workers by Different Occupations and Industries: A Longitudinal Population-Based Matched Case-Control Study in Taiwan”
Point#2: In the introduction, you cite 3 possible explanations for the relationship between BC and occupational categories: Chemical exposure, Shift work (disrupting diurnal rhythm), Sedentary occupations. You also follow these up when you are discussing your results. However, you also throw in a number of other factors that are unsubstantiated as you did not control for these. A notable example is 14: 294-296 where you claim that: “higher breast cancer risk in some white-collar occupations, such as legal and accounting activities, education, and medicine, possibly partly associated with reproductive risk factors including older age at first childbirth and less parity due to longer training time”. This needs further justification with reference to related research.
Our response: We appreciate the Reviewer’s suggestion. We revised and cited reference for the paragraph as follows:
“We observed higher breast cancer risk in some white-collar occupations, such as legal and accounting activities, education, and medicine, possibly partly associated with reproductive risk factors including older age at first child birth and less parity due to longer training time in highly professional [38,39]”
Reference:
(38) Kullberg C, Selander J, Albin M, Borgquist S, Manjer J, Gustavsson P. Female white-collar workers remain at higher risk of breast cancer after adjustments for individual risk factors related to reproduction and lifestyle. Occupational and environmental medicine. 2017;74(9):652-658.
(39) Butt S, Borgquist S, Anagnostaki L, Landberg G, Manjer J. Parity and age at first childbirth in relation to the risk of different breast cancer subgroups. International journal of cancer. 2009;125(8):1926-1934.
Point#3: Your findings are not always effectively related to other research studies. For instance, 12:202-204 you cite Ji et al., 2008 but don’t relate this to your findings. Sometimes your extensive citation of other studies reads more like a list rather than as part of an interpretative analysis. It may be an idea to consider just focusing on Asian studies with a separate section to cover how these findings compare with Westernised evidence for an association between occupational sub-group and breast cancer incidence. You may also want to thi nk about tabulating some of the known occupational risk factors as this may break up the density of your background text.
Our response: We appreciate the Reviewer’s suggestion. Regarding the statements cited from Ji et al., 2008, we want to present there were similar findings, which investigated the relation between breast cancer and whole industries in the existing international studies. We revised the paragraphs and as follows:
“Some of our findings were consistent as those from previous population-based international research investigating the relation between breast cancer and kinds of occupations. For example, Ji et al. (2008) conducted a population-based cohort study in Shanghai and reported higher breast cancer risk associated with several production industries, postal/telecommunication workers and white-collar professionals. Pukkala et al. (2009) investigated the relation between occupation and cancer among 15 million people in five Nordic countries and found higher breast cancer risk among jobs averagely required high-level education and lower risk among those jobs physically demanded.”
Point#4: The claim in your conclusion that the occupational groups with highest levels of BC would benefit from education about early detection is not appropriate or sufficiently justified and referenced. Do you mean education to enhance awareness of early detection (i.e., symptom awareness, BSE/mammography)? / Are you referring to the need for awareness campaigns about possible risk factors associated with certain occupations? In either case, you need to contextualise this conclusion in more detail throughout the rest of the paper. As it stands, it doesn’t seem to be a logically informed or clearly expressed conclusion to the paper.
Our response: We appreciate the Reviewer’s suggestion. We revised conclusion section as follows:
“As recognized the risks, policy intervention of occupational health may be necessary to optimize strategies to prevent breast cancer risks among female workers. Regular mammography check may be needed to be include in the annual labor physical examination for those high-risk occupations.”
Reviewer 2 Report
Comments and suggestions for authors
International Journal of Environmental Research and Public Health
Manuscript number: ijerph-1821045
Title: Breast Cancer Risks among Female Workers by Different Occupations and Industries: A Longitudinal Population-Based Matched Case-Control Study in Taiwan
General
This study by Shen et. al., was planned to investigate breast cancer incidence among female workers from different occupations; specifically, they conducted a case-control study among women with an incident breast cancer diagnosis, matched with those without breast cancer. Using three nationwide population-based labor insurance databases in Taiwan to explore potential breast cancer risks in different occupational industries over a lifetime period.
Major Concerns
I really appreciate that the authors mention the limitations of the study, among them, the unobservable risk factors including hormone-related risk factors, family history, alcohol consumption and possible occupational exposure. If well they do not have access to all that information, I think is necessary to do a correlation between the risk factors to which the workers of that occupations are exposed; since in the discussion section they only write a possible explanation, but the impact of this investigation will increases if they include this additional analysis. If the authors have not access to other parameters how they can ensure that the risk factor increases due the occupation.
I suggest that the authors mention in the abstract and conclusion sections the significant breast cancer risks with higher OR, such as manufacturing; including, manufacture of tobacco products (OR=2.5), manufacture of petroleum and coal products (OR=1.633), manufacture of electronic parts and components (OR=1.306), manufacture of electrical equipment (OR= 1.309). Since reading the abstract and conclusion it appears that all occupations increases the risk of developing breast cancer, rather than occupations, could be the associated and classified risk factors, which are missing.
Minor Concerns
I suggest to divide the Table 2 in two parts, one of them could include the OR values below one and the other part could include the OR values higher than one.
The authors excluded the total cumulative job tenure less than 3 months in working industries; however, In the group that developed breast cancer, what was the length of time that the woman were working in that job before de diagnosis, is to say how many time they were exposed to that job.
Establish the same format in the tables, for example in the Table I, age categories, the words between parentheses are separated by comma “(N, %); however in major working locations is written as “(N%), I suggest also separate by comma.
Table 1: Total cumulative job tenures in years (mean ± SD), it should be in the parentheses (N, %), since they are no showing the mean of years nor standard deviation.
Table 1. What percentage of women in every job tenure category in years show breast cancer?.
Table 2: I suggest increasing the width of the cells, since result difficult to read. Is to say, that all the headings and values appear in only one line.
Table 2: In the classification system Electricity and gas supply and Education there are only a unique category, therefore the data should be the same; however the OR value are different.
Author Response
This study by Shen et. al., was planned to investigate breast cancer incidence among female workers from different occupations; specifically, they conducted a case-control study among women with an incident breast cancer diagnosis, matched with those without breast cancer. Using three nationwide population-based labor insurance databases in Taiwan to explore potential breast cancer risks in different occupational industries over a lifetime period.
Point#1: I really appreciate that the authors mention the limitations of the study, among them, the unobservable risk factors including hormone-related risk factors, family history, alcohol consumption and possible occupational exposure. If well they do not have access to all that information, I think is necessary to do a correlation between the risk factors to which the workers of that occupations are exposed; since in the discussion section they only write a possible explanation, but the impact of this investigation will increases if they include this additional analysis. If the authors have not access to other parameters how they can ensure that the risk factor increases due the occupation.
Our response:
We appreciate the reviewer’s suggestion. Our study used the secondary data analysis which could not directly estimate possible risk factors and could just discussed the possible exposure in different section and division level of the industries. We revised the limitation and as follows:
“However, our study had some limitations. First, several unobservable risk factors could be confounders because of secondary data analysis, including hormone-related risk factors, family history, alcohol consumption, and possible occupational exposures such as night shift work, organic chemicals, physical activities, working hours and even different exposure and its duration in each section and division level of the industry. We also lack the records in labors’ attendance including the information of sick leaves for maternity or pregnancies in each industry. Second, some subgroup occupations with higher breast cancer risk may be overestimated due to smaller sample sizes. Third, the total cumulative job tenures were as being confounding factors for conditional logistic regression in our study. However, we did not particularly investigate the association between breast cancer incidence and job tenures even each job employment duration and exposure time before breast cancer diagnosis in each industry. Further study may investigate the relation between breast cancer incidence and each job employment duration in each industry. Finally, our findings may not generalize to other countries because the occupational exposures may be at different levels in each country.”
Point#2: I suggest that the authors mention in the abstract and conclusion sections the significant breast cancer risks with higher OR, such as manufacturing; including, manufacture of tobacco products (OR=2.5), manufacture of petroleum and coal products (OR=1.633), manufacture of electronic parts and components (OR=1.306), manufacture of electrical equipment (OR= 1.309). Since reading the abstract and conclusion it appears that all occupations increases the risk of developing breast cancer, rather than occupations, could be the associated and classified risk factors, which are missing.
Our response:
We appreciate the reviewer’s suggestion. This study conducted an ecological study and investigated the breast cancer incidence by different section and division levels according to standard industrial classification of Taiwan’s National Statistics. In general, there are both advantages and disadvantages when using large longitudinal database to conduct such kinds of study. Specifically, given the study had limit information of possible exposures, our findings presented the general epidemiology evidences in the association between breast cancer incidence and possible industries.
Point#3: I suggest to divide the Table 2 in two parts, one of them could include the OR values below one and the other part could include the OR values higher than one.
Our response: We appreciate the reviewer’s suggestion. The industrial section and division levels were based on the standard industrial classification of Taiwan’s National Statistics. To be able to easily interpret results for future governmental occupational health policy planning, we reshape the results into Table 2 at the section level in the revised version, and the division level in the supplementary Appendix eTable1.
Point#4: The authors excluded the total cumulative job tenure less than 3 months in working industries; however, In the group that developed breast cancer, what was the length of time that the woman were working in that job before de diagnosis, is to say how many time they were exposed to that job.
Our response: We appreciate the reviewer’s suggestion. We excluded those patients with the total cumulative job tenure less than 3 months who were thought as less potential exposure of breast cancer risk in the working space. Our study only calculated total cumulative job tenures and controlled for conditional logistic regression and did not further calculate the job employment duration of each industry for individual women, which was one of the limitations in our study. We revised the part of limitation and as follows:
“Second, some subgroup occupations with higher breast cancer risk may be overestimated due to smaller sample sizes. Third, the total cumulative job tenures were as being confounding factors for conditional logistic regression in our study. However, we did not particularly investigate the association between breast cancer incidence and job employment duration of each industry for individual women. Further study may investigate the relation between breast cancer incidence and each job employment duration in each industry. Finally, our findings may not generalize to other countries because the occupational exposures may be at different levels in each country.”
Point#5: Establish the same format in the tables, for example in the Table I, age categories, the words between parentheses are separated by comma “(N, %); however in major working locations is written as “(N%), I suggest also separate by comma.
Our response: We appreciate the reviewer’s suggestion. We revised (N%) in major working locations as (N, %). Please find the attached completed memo.
Point#6: Table 1: Total cumulative job tenures in years (mean ± SD), it should be in the parentheses (N, %), since they are no showing the mean of years nor standard deviation.
Our response: We appreciate the reviewer’s suggestion. We revised (mean ± SD) in Total cumulative job tenures categories in years as (N, %) . Please find the attached completed response memo.
Point#7: Table 1. What percentage of women in every job tenure category in years show breast cancer?.
Our response: We appreciate the reviewer’s suggestion. We conducted a case-control study to include women with incident diagnosed as breast cancer between 2008 and 2017 and compare with matched cohorts without any breast cancer for their past life-time working occupations and calculated cumulative job tenure in working industries between 1950 and 2017. As the Table 1 shows, the mean age of incident breast cancer were 53.21 years and mean cumulative job tenures were 15.97 years among women with incident breast cancer. The percentage of women in every job tenure in years were 3.49% for less than 1 year, 6.79% for 1-3 year, 6.43% for 3-5 years, 15.56% for 5-10 years, 29.92% for 10-20 years and 37.81% for over 20 years. We added more descriptions in the result section as follows:
“Table 1 summarizes demographic characteristics, working locations, and total cumulative job tenure between matched case and control female workers based on their index age and working locations. Mean index ages for matched cohorts were 53.21 years, and greater proportions worked in the major working locations of Taipei, Taichung, and Kaohsiung. Mean total cumulative job tenures were 15.97 years among women with incident breast cancer. The percentage of women in every job tenure in years were 3.49% for less than 1 year, 6.79% for 1-3 year, 6.43% for 3-5 years, 15.56% for 5-10 years, 29.92% for 10-20 years and 37.81% for over 20 years.”
Point#8: Table 2: I suggest increasing the width of the cells, since result difficult to read. Is to say, that all the headings and values appear in only one line.
Our response: We appreciate the reviewer’s suggestion and reformatted Table 2 at the section level and the division level in the supplementary Appendix eTable1.
Point#9: Table 2: In the classification system Electricity and gas supply and Education there are only a unique category, therefore the data should be the same; however the OR value are different.
Our response: We appreciate the reviewer’s comment. We analyzed conditional logistic regression and controlling for the confounding variables separately for the section level and division level variables. Therefore, the OR value would be different. To clarify and reduce confusion, in the revised version, we reshaped results at the section level in the Table 2 and moved results at division level in the Appendix eTable1.
Reviewer 3 Report
Overall, this is a well written paper that used a case control design to evaluate occupations and their association with breast cancer development.
Recommendations:
1) While the conclusions are supported by the results, the overall greater risk amongst all occupations was rather small, with the greatest risk being for individuals in education. Even the risk in education, however, is rather low, with an OR of only 1.199. I recommend adding a statement to qualify that these patients are only at a slight increased risk to not overstate the conclusions.
2) Sentence on lines 83-84, remove the word however for clarity.
3) Lines 95-100 should be moved to the first part of the methods section.
4) Figure 1: Some statements have punctuation at the end, while others do not. Please make consistent. Also, working is spelled incorrectly in the study case group second box. There are also space inconsistencies in the comparison group second box.
5) Both p-value and P value are used throughout the paper/tables. Please make consistent.
6) Line 177, I recommend adding the outcome of interest to this line for readability. For example: ..., significantly greater percentages of case groups (patients with breast cancer) worked in manufacturing...
7) Line 178 - It states that manufacturing is section B, however, it is section C in the table
8) Table 2 - please correct formatting. Additionally, while it has a lot of interesting information presented, it is quite overwhelming. I recommend submitting the entire table as a supplemental table and modifying the table in the text to only include the overall category results.
9) As many breast cancers are diagnosed via screening, and in many countries, screening adherence varies with occupation as well (for multiple reasons, including socioeconomic status, educational level, insurance coverage), can the authors please comment on the typical screening adherence in Taiwan and whether that could potentially influence the results as well?
Author Response
Point#1: While the conclusions are supported by the results, the overall greater risk amongst all occupations was rather small, with the greatest risk being for individuals in education. Even the risk in education, however, is rather low, with an OR of only 1.199. I recommend adding a statement to qualify that these patients are only at a slight increased risk to not overstate the conclusions.
Our response: We appreciate the reviewer’s suggestion. We revised the abstract and conclusion as these patients are only with slight increased breast cancer risk, and as follows:
“Conclusions: Greater percentages of industrial occupations (i.e., manufacturing, wholesale and retail, or health professionals) were associated with slightly increased breast cancer risk. As recognized the risks, policy intervention of occupational health and regular mammography check may be necessary to prevent breast cancer risks among female workers. “ (in the abstraction section)
“Our study found slightly significant breast cancer risk among the major classification of occupations including manufacturing, wholesale and retail trade, information and communication activities, financial and insurance activities, real estate activities, professions, scientific and technical activities, public administration, defense, social security, education, and human health and social work activities from the section and division levels of standard industrial classification of Taiwan’s National Statistics. As recognized the risks, policy intervention of occupational health may be necessary to optimize strategies to prevent breast cancer risks among female workers. Regular mammography check may be needed to be include in the annual labor physical examination for those high-risk occupations. “ (in the conclusion section)
Point#2: Sentence on lines 83-84, remove the word however for clarity.
Our response: We appreciate the reviewer’s correction. We revised the sentence from introduction and as follows: “Most previous studies were conducted in Western countries, however, with fewer population-based studies focused on Asia.”
Point#3: Lines 95-100 should be moved to the first part of the methods section.
Our response: We appreciate the reviewer’s suggestion. We revised Lines 95-100 from the introduction and moved to the first part of methods as follows:
“Introduction: In Taiwan, studies focused on breast cancer risk among different occupations remain limited, especially studies with longer follow-up time. This study aimed to investigate breast cancer incidence among female workers from different occupations. Specifically, we conducted a case-control study among women with an incident breast cancer diagnosis between 2008 and 2017, matched with those with no breast cancer diagnosis. Using three nationwide population-based labor insurance databases in Taiwan, we then tracked the study cohorts’ labor enrollment records for almost 70 years from 1950 to 2017 to explore potential breast cancer risks in different occupational industries over a lifetime period.
Method
2.1 Study design and data source
This study conducted a retrospective case-control study design among women with an incident breast cancer diagnosis between 2008 and 2017, matched with those with no breast cancer diagnosis. Using three nationwide population-based labor insurance databases in Taiwan, we then tracked the study cohorts’ labor enrollment records for almost 70 years from 1950 to 2017 to explore potential breast cancer risks in different occupational industries over a lifetime period. The first was the National Labor Insurance Database (NLID) from the Ministry of Labor, containing representative data on labor enrollment profiles from 1950 to 2017. “
Point#4: Figure 1: Some statements have punctuation at the end, while others do not. Please make consistent. Also, working is spelled incorrectly in the study case group second box. There are also space inconsistencies in the comparison group second box.
Our response: We appreciate the reviewer’s correction. We revised and deleted the punctuation at the end of the statements, corrected the spells and space consistencies. The revised Figure1 was presented and as follows:
Point#5: Both p-value and P value are used throughout the paper/tables. Please make consistent.
Our response: We appreciate the reviewer’s suggestion. We revised p-value from the tables as P value from the tables
Point#6: Line 177, I recommend adding the outcome of interest to this line for readability. For example: ..., significantly greater percentages of case groups (patients with breast cancer) worked in manufacturing...
Our response: We appreciate the reviewer’s suggestion. We revised the sentence and as follows:
“ Table 2 provides ORs, and multiple testing results by the standard industrial classification system at the section and division levels. Compared with comparison groups at the section level, significantly greater percentages of case groups (patients with breast cancer) worked in manufacturing (section C; 69.85%, OR 1.027, P 0.0002);”
Point#7: Line 178 - It states that manufacturing is section B, however, it is section C in the table
Our response: We appreciate the reviewer’s suggestion. We revised the sentence and as follows:
“ Table 2 provides ORs, and multiple testing results by the standard industrial classification system at the section and division levels. Compared with comparison groups at the section level, significantly greater percentages of case groups (patients with breast cancer) worked in manufacturing (section C; 69.85%, OR 1.027, P 0.0002);”
Point#8: Table 2 - please correct formatting. Additionally, while it has a lot of interesting information presented, it is quite overwhelming. I recommend submitting the entire table as a supplemental table and modifying the table in the text to only include the overall category results.
Our response: We appreciate the reviewer’s suggestion and reformatted Table 2 at the section level and the division level in the supplementary Appendix eTable1.
Point#9: As many breast cancers are diagnosed via screening, and in many countries, screening adherence varies with occupation as well (for multiple reasons, including socioeconomic status, educational level, insurance coverage), can the authors please comment on the typical screening adherence in Taiwan and whether that could potentially influence the results as well?
Our response: We appreciate the reviewer’s suggestion. Our study did not further examine mammography rate among occupational industries. Future study may further investigate whether mammography rate varies with occupations. We addressed this in the limitation section as follows:
“Fourth, mammography screening utilization and screening adherence might be also a potential confounder while investigating breast cancer risks among these occupational industries. However, the current study did not examine mammography rate among occupational industries. Future study may further investigate whether mammography rate varies with occupations.”
Reviewer 4 Report
The main objective of this article is to identify the industries at risk for breast cancer in Taiwan
The main interest of this work resides in the fact that few data are available on the risk of breast cancer in Asia, and even less in Taiwan
However, this work has methodological limitations that reduce its scientific impact
The main limitation is the absence of information on the job occupied, only the sector of activity being indicated. Yet exposures will be very different depending on the job performed within the same sector of activity. Moreover, the duration of exposure in the activity sector was not considered. This could probably be done on the basis of available data.
The second major limitation is the lack of information on general risk factors for breast cancer, which are well described by the authors in the introduction. Unfortunately, it seems that this information on risk factors is not available for this study. Therefore, would it be possible to have an indirect estimate of these factors? Do you have proxies to assess this: information on sick leaves for maternity, statistics on the number of pregnancies by industrial sector...
Minor comments
1) I suggest not mentioning ionizing radiation as an occupational risk for breast cancer because the IARC classification for this site for radiation was based on medical exposure.
2) All the subjects with previous cancer diagnosis were excluded from the analysis. What is the rational for this? I suggest performing sensitivity analysis without exclusion to assess the impact of this choice on the results obtained.
3) The authors mentioned that de conditional logistic regression was controlled for total cumulative job tenure years. What is the rational for this? I believe it would be better to perform complementary analyses by employment duration e.g more than 2 or 5 or 10 years in each industry considered.
4) In the conclusion section, it is mentioned that the study found increased risk of breast cancer among the major classification of occupations. However the analysis did not consider occupation but industrial sector of activity of the company. Please use the appropriate terminology.
Author Response
Point#1: The main limitation is the absence of information on the job occupied, only the sector of activity being indicated. Yet exposures will be very different depending on the job performed within the same sector of activity. Moreover, the duration of exposure in the activity sector was not considered. This could probably be done on the basis of available data.
Our response: We appreciate the reviewer’s comment. Our study used the secondary data analysis which could not direct estimate possible risk factors and even lack the information including the possible exposure and duration in each industry. We revised the part of limitation and as follows:
“However, our study had some limitations. First, several unobservable risk factors could be confounders because of secondary data analysis, including hormone-related risk factors, family history, alcohol consumption, and possible occupational exposures such as night shift work, organic chemicals, physical activities, working hours and even different exposure and its duration in each section and division level of the industry. We also lack the records in labors’ attendance including the information of sick leaves for maternity or pregnancies in each industry.”
Point#2: The second major limitation is the lack of information on general risk factors for breast cancer, which are well described by the authors in the introduction. Unfortunately, it seems that this information on risk factors is not available for this study. Therefore, would it be possible to have an indirect estimate of these factors? Do you have proxies to assess this: information on sick leaves for maternity, statistics on the number of pregnancies by industrial sector...
Our response: We appreciate the reviewer’s suggestion. Our study used the secondary data analysis which could not direct estimate possible risk factors and even lack the information including sick leaves for maternity or pregnancies from the workers in each industry. We revised the part of limitation and as follows:
“However, our study had some limitations. First, several unobservable risk factors could be confounders because of secondary data analysis, including hormone-related risk factors, family history, alcohol consumption, and possible occupational exposures such as night shift work, organic chemicals, physical activities, working hours and even different exposure and its duration in each section and division level of the industry. We also lack the records in labors’ attendance including the information of sick leaves for maternity or pregnancies in each industry.”
Point#3: I suggest not mentioning ionizing radiation as an occupational risk for breast cancer because the IARC classification for this site for radiation was based on medical exposure.
Our response: We appreciate the reviewer’s suggestion. We revised and removed ionizing radiation from the introduction, and as follows:
“The International Agency for Research on Cancer (IARC) noted several factors with sufficient or limited evidence for human breast cancer risk, including X- and gamma-radiation, night-shift work, dieldrin, ethylene oxide, and polychlorinated biphenyls. Ionizing radiation had a linear dose–response relationship with breast cancer incidence, with greater risk if exposure occurred before age 20 years.”
Point#4: All the subjects with previous cancer diagnosis were excluded from the analysis. What is the rational for this? I suggest performing sensitivity analysis without exclusion to assess the impact of this choice on the results obtained.
Our response: We appreciate the reviewer’s suggestion. We conducted a case-control study to include women with incident diagnosed as breast cancer between 2008 and 2017 and compare with matched cohorts without any breast cancer for their past life-time working occupations and calculated cumulative job tenure in working industries between 1950 and 2017. In addition, to reduce potential bias due to prior cancer history, we excluded patients with previously cancer diagnosis from both case and comparison groups to have make both groups comparable.
Point#5: The authors mentioned that de conditional logistic regression was controlled for total cumulative job tenure years. What is the rational for this? I believe it would be better to perform complementary analyses by employment duration e.g more than 2 or 5 or 10 years in each industry considered.
Our response: We appreciate the reviewer’s suggestion. Our study only calculated total cumulative job tenures controlled for conditional logistic regression. The job employment duration of each industry was one of the limitations in our study. Further study may investigate the possible risk factors adjusting each job employment duration in each industry. We revised the part of limitation and as follows:
“Third, the total cumulative job tenures were as being confounding factors for conditional logistic regression in our study. However, we did not particularly investigate the association between breast cancer incidence and job employment duration of each industry for individual women. Further study may investigate the relation between breast cancer incidence and each job employment duration in each industry.”
Point#6: In the conclusion section, it is mentioned that the study found increased risk of breast cancer among the major classification of occupations. However, the analysis did not consider occupation but industrial sector of activity of the company. Please use the appropriate terminology.
Our response: We appreciate the reviewer’s suggestion. We revised the conclusion and as follows:
“Our study found slightly significant breast cancer risk among the major classification of occupations including manufacturing, wholesale and retail trade, information and communication activities, financial and insurance activities, real estate activities, professions, scientific and technical activities, public administration, defense, social security, education, and human health and social work activities from the section and division levels according to the standard industrial classification of Taiwan’s National Statistics. As recognized the risks, policy intervention of occupational health may be necessary to optimize strategies to prevent breast cancer risks among female workers. Regular mammography check may be needed to be include in the annual labor physical examination for those high-risk occupations.”
Round 2
Reviewer 1 Report
You have effectively revised and improved the manuscript. This paper will be a significant addition to the field of occupational breast cancer incidence. Well done!
Author Response
Point#1: You have effectively revised and improved the manuscript. This paper will be a significant addition to the field of occupational breast cancer incidence. Well done!
Our response: We appreciate the Reviewer’s comment and encouragement.

Reviewer 2 Report
I thank the authors for the answer to each point and the changes made to the manuscript in order to improve it.
Author Response
Point#1: I thank the authors for the answer to each point and the changes made to the manuscript in order to improve it.
Our response: We appreciate the Reviewer’s comment and suggestion.

Reviewer 4 Report
Thank you for your answers to the questions raised.
I disagree with the last sentence of your conclusion. Your results do not allow for such a medical follow-up recommendation.
Author Response
Reviewer#4
Point#1: I disagree with the last sentence of your conclusion. Your results do not allow for such a medical follow-up recommendation.
Our response: We appreciate the Reviewer’s suggestion. We agreed and revised the conclusion as follows:
“Conclusion: Our study found slightly significant breast cancer risk among the major classification of occupations including manufacturing, wholesale and retail trade, information and communication activities, financial and insurance activities, real estate activities, professions, scientific and technical activities, public administration, defense, social security, education, and human health and social work activities from the section and division levels according to the standard industrial classification of Taiwan’s National Statistics. As recognized the risks, policy intervention of occupational health may be necessary to optimize strategies to prevent breast cancer risks among female workers. Regular mammography check may be needed to be include in the annual labor physical examination for those high-risk occupations. Further study should investigate the possible exposure factors among female workers in those industries with slightly higher incident breast cancer. “
